# The Historical Development of Cultivation Techniques for Methanogens and Other Strict Anaerobes and Their Application in Modern Microbiology

**DOI:** 10.3390/microorganisms10020412

**Published:** 2022-02-10

**Authors:** Nikola Hanišáková, Monika Vítězová, Simon K. -M. R. Rittmann

**Affiliations:** 1Laboratory of Anaerobic Microorganisms, Section of Microbiology, Department of Experimental Biology, Faculty of Science, Masaryk University, 62500 Brno, Czech Republic; 446645@mail.muni.cz; 2Archaea Physiology & Biotechnology Group, Department of Functional and Evolutionary Ecology, Universität Wien, 1030 Wien, Austria

**Keywords:** methane, anaerobes, methanogens, biogas, cultivation methods

## Abstract

The cultivation and investigation of strictly anaerobic microorganisms belong to the fields of anaerobic microbial physiology, microbiology, and biotechnology. Anaerobic cultivation methods differ from classic microbiological techniques in several aspects. The requirement for special instruments, which are designed to prevent the contact of the specimen with air/molecular oxygen by different means of manipulation, makes this field more challenging for general research compared to working with aerobic microorganisms. Anaerobic microbiological methods are required for many purposes, such as for the isolation and characterization of new species and their physiological examination, as well as for anaerobic biotechnological applications or medical indications. This review presents the historical development of methods for the cultivation of strictly anaerobic microorganisms focusing on methanogenic archaea, anaerobic cultivation methods that are still widely used today, novel methods for anaerobic cultivation, and almost forgotten, but still relevant, techniques.

## 1. Introduction

In the field of microbiology, anaerobic cultivation methods differ from the classic methods for aerobic cultivation. The main reason for the difference is that while manipulating strictly anaerobic microorganisms, it is necessary to prevent molecular oxygen (O_2_) exposure to the organisms. This is due to the fact that O_2_ is toxic to anaerobic microorganisms, to varying degrees and depending on the microbe [1], and on their oxidation-reduction potential (ORP), the optimal value of which differs among anaerobic species. As awareness of microorganisms that are not capable of growing while exposed to air was increasing towards the end of the 19th century, the need for special methods for their cultivation and manipulation was realized. The first efforts began more than a century ago and continued until the present variety of anaerobic cultivation methods and procedures was established [2,3,4,5]. For the most common anaerobic and aerotolerant microorganisms, cultivation techniques are often manageable under ordinary laboratory conditions, but more sophisticated cultivation methods are required for the study of strictly anaerobic microorganisms, such as methanogens, whose importance and biotechnological applications are now of utmost interest [6,7,8,9].

The relatively demanding procedures for handling strictly anaerobic microorganisms and technical requirements often prevent the cultivation of these organisms in common microbiological laboratories. Apart from this, there are laboratory limitations towards many different anaerobic cultivation setups with regard to experimental conditions. That means using high-throughput methods or studying requirements towards a particular gas composition and applied pressure. In addition, the toxicity and flammability of some microbial substrates and product gasses alone is an important factor, which might render cultivation difficult or impossible in standard microbiological laboratories.

It is of importance to know methods of anaerobic cultivation to perform such experiments, study microorganisms responsible for anaerobic substrate conversion, and measure their metabolic activity and degradation potential to enable us to fully understand their physiological processes. Mastering anaerobic cultivation techniques is necessary to describe new species, for anaerobic research and development, and to advance anaerobic microbiology and biotechnology.

The aim of this work is to present methods of anaerobic cultivation and their historical development. Although the primary focus of these techniques is to capture broad methods developed for cultivation of methanogenic archaea (methanogens), they are applicable to other strictly anaerobic microorganisms, for example clostridia, sulphate-reducing bacteria (SRB), or phototrophic bacteria. In addition, this review will introduce the methods of anaerobic microbiology that have been almost forgotten or are no longer in use, and show their potential applicability in modern microbiology and biotechnology. This insight into the principles of anaerobic cultivation might help scientists to choose among these methods to successfully isolate, cultivate and perform biotechnological experiments with anaerobic organisms.

## 2. Milestones in the Historical Development of Anaerobic Microbiology

### 2.1. The Origin of Anaerobic Cultivation

Methods for cultivation of microorganisms have been developed and adapted since their discovery. Initially, the studies were aimed at discovering the connection of microbes to infection, illnesses and to biotechnological applications [10,11]. Understanding the cause of the infections and diseases was crucial as it would lead to an understanding of their transmission, prevention and treatment.

Since the first discovery of ‘animalcules’ in 1680 in Leeuwenhoek’s specimen, the awareness of organisms smaller than the human eye can see brought a new perspective on how the world and its processes might be dependent on microbes. Even then, the anaerobic microorganisms noticed for the first time as some of the ‘animacules’ were growing in sealed glass tubes and producing gas [12]. The existence of organisms that could not be seen by eye was, of course, something that was never heard of before. However, the biggest development in microbial cultivation techniques occurred in the 20th century, when the importance of the natural processes caused by microorganisms increased [13,14,15,16].

It is understandable that the first cultivation techniques were aerobic and led to the discovery and description of bacteria. The greatest advance in the field of microbial cultivation was made by Robert Koch, who used prepared media for cultivation. The media were made from broth based on boiled meat or bovine serum. He also used media solidified by gelatine. However, working with gelatine was also unfruitful, as some bacterial species are capable of degrading gelatine, and at temperatures higher than 30 °C it is impossible to achieve gelatine solidification. Walter Hesse, with his wife’s suggestion, replaced the gelatine with agar. Gelatin-like agar is harvested from the red algae *Gelidium* sp. and *Gracilaria* sp. (division *Rhodophyta*). Agar is more stable, a dilute solution solidifies at 37 °C, does not affect the growth, and is not degraded by most of the microorganisms [17]. It was a breakthrough for the cultivation of plateable organisms.

Anaerobic microorganisms came to Pasteur’s attention during his research on the acidification of wine. The process was caused by contamination of bacteria that led to the production of acids. He described the fermentation process and suggested that the wine-making process was brought about by anaerobic microorganisms. Pasteur also described the effect that the substrate uptake rate increases under anaerobic conditions compared to aerobic conditions to produce the same amount of biomass, the so-called Pasteur effect. He cultivated *Clostridium butyricum* using meat extract broth as the medium, which was introduced to a vacuum. Under close examination of the microorganisms under the microscope, he found that the exposure of the cells to O_2_ resulted in a decrease in motility in some cases. He used the terms ‘aerobic’ and ‘anaerobic’ for the first time and also developed the first techniques of anoxic media preparation. These involved boiling or application of a vacuum [18,19].

The recognised significance of anaerobic microorganisms grew over the years, as they were discovered to be the cause of intoxications and infections leading to gas gangrenes [4,11,20,21]. Studies of the infections led to an increase in the number of anaerobes isolated and described. In order to cultivate these anaerobes, modifications of the basic microbiological methods needed to be carried out, and the number of reviews on that topic at that time also increased. In 1929, I. C. Hall extended his description of anaerobic bacteria isolation [2] and summarized the known methods of anaerobic cultivation in one of his reviews [13]. These methods later led to the isolation and cultivation of *Clostridioides difficile*. Thereafter, an interest in the study of *C. difficile* emerged, as the organism had been described as an anaerobic, pathogenic microorganism [22]. A typical method for anaerobic cultivation used deep tube techniques, with the medium in a semisolid or solid state (Table A1). For removing O_2_, boiling of the medium was recommended. The depth of the media column ensured that O_2_ reabsorption occurred only in a small surface volume and anaerobiosis persisted at the bottom of the tube. To reduce the contact with air, constricted tubes came to use, improved with a mechanical marble seal (Figure 1) [23]. Another option is to apply a layer of mineral oil that seals the surface of the liquid medium and eliminates contact with air. Since there are many types of oil, Hall suggests that paraffin oil is the most sufficient to serve as seal [13].

Another anaerobic cultivation method is Buchner’s method, in which pyrogallic acid is added onto the cotton plug inside the tube and the tube is then sealed with a rubber plug [5]. The method was later modified [24,25,26]. In addition, gas exhaustion and the use of inert gas were also described one hundred years ago as a form of anaerobic cultivation [4,21,27]. During experiments with fermentation and anaerobic cultivation of mixed samples, gas was produced, which aroused curiosity concerning the composition of this gas. The main components produced from selected anaerobic species were described as carbon dioxide (CO_2_), molecular hydrogen (H_2_) [28], and traces of hydrogen sulphide (H_2_S) [29]. Denitrification by bacteria was described as well as its inhibition by O_2_ [30,31].

In the mid-20th century, interest in anaerobic cultivation increased, as it was discovered that a large number of microorganisms possess strict anaerobic cultivation requirements in contrast to the majority of species that had been anaerobically cultivated until that time. Moreover, some of these anaerobic microorganisms were found to be responsible for cellulose degradation in the digestive tract of ruminants, or they were isolated from anaerobic infections in humans [10].

Efforts to cultivate these strictly O_2_-sensitive organisms resulted in the further modification of anaerobic culturing techniques and prepared the space for the cultivation of methanogens. The focus was on the cultivation of anaerobic bacteria from the rumen environment, such as cellulose and xylene-degrading bacteria [32,33]. A pioneer in the further development of anaerobic cultivation techniques was Hungate. His studies on cellulolytic bacteria led to the isolation of species such as *Fibrobacter succinogenes* (formerly *Ruminobacter succinogenes*) [33].

### 2.2. First Attempts of Methanogens Isolation and the Discovery of Archaea

In 1776, Alessandro Volta collected the gas formed bubbling in swamps [3,34]. He discovered that the gas was flammable. Almost a hundred years later, the gas was named methane by the German chemist August Wilhelm von Hofmann [35]. Further observations and gas analyses revealed that the gas formed in swamps consists of methane, CO_2_ and N_2_. Thereafter, the microbiological role in methane formation started to be studied in more detail.

Initially, methane formation and cellulose degradation were often associated with each other, and cellulose decomposition and gas formation were deeply studied [36,37]. The presence of methane in the formed gas led to studies based on two topics. The first was aimed at the isolation of methanogens [3,38], and second was the discovery of substrates used by these methanogens [39].

At the turn of the 20th century, Omelianski studied the fermentation of cellulose and noticed that some microorganisms produced H_2_ and some methane [36]. He also distinguished them based on temperature surveillance [3,36]. The methane bacilli were killed by pasteurisation while the spores of the H_2_-producing bacilli survived, and using the heating procedure, he obtained only the H_2_-producing consortium. In 1915, Mazé found spherical microorganisms in his enrichment cultures that produced methane. They fermented acetate to methane and formed aggregates [28,38]. In 1910, Söhngen discovered two types of methanogens. He described methanogenic rods and also observed and isolated acetate-utilizing methanogens that formed cell bundles and named them *Sarcina* (*sarcina* (lat.) meaning bundle) [3]. Söhngen’s experiments also proved that added H_2_ is rapidly consumed by the microorganisms and leads to methane production.

Based on his observations and calculations, he confirmed Equation (1) proposed by Omelianski:CO_2_ + 4H_2_ → CH_4_ + 2H_2_O(1)

His observations were published in his work with photographs of these microorganisms, coining the genera *Methanobacterium* and *Methanosarcina* [3].

The isolation of pure cultures of methanogens is complicated due to their strictly anaerobic nature. Agar plates could not be used, as the microorganisms did not grow on them because they were not incubated in a strictly anaerobic atmosphere. Agar shake tubes, however, proved efficient for isolating colonies.

The first methanogenic pure culture isolated was *Methanobacterium omelianski* by Barker in 1936 [38]. In the same work, he described the isolation and description of Söhngen’s methane-producing bacterium and named it *Methanobacterium söhngenii* (now *Methanothrix söhngenii*). In that work, he also described the isolation of Söhngen’s *Sarcina*. The isolated strain was able to form methane only from calcium acetate and was named *Methanosarcina methanica*. He also succeeded in isolating Mazé’s cocci and named them *Methanococcus mazei* (now *Methanosarcina mazei*). He emphasised that the latter three species were not strictly pure, but were only purified by repeated transfers and applications of techniques as a combination of agar shake tubes and special Hall’s tubes with marble seals.

Only later, the supposedly pure *Methanobacterium omelianski*, capable of oxidation of ethanol and further described in Barker’s next work [40], was discovered to consist of two different species. These were *Methanobacterium bryantii* and the “S” organism, which oxidized ethanol to acetic acid and produced H_2_ that served as a substrate for the syntrophically growing methanogen [41,42]. Other species of methanogens that were isolated were *Methanococcus vannielii* [43] and *Methanobrevibacter ruminantium* [44].

The breakthrough in anaerobic cultivation was achieved by Hungate. He innovated his techniques for cultivation of cellulolytic microorganisms, which led to new discoveries in the field of microbiology. In 1969, Hungate published “A Roll Tube Method for Cultivation of Strict Anaerobes”, which has become the most important source of techniques and procedures for anaerobic cultivation. Many of the principles described by Hungate or derivatives of these methods are widely used by many scientists today [14]. The application of these modifications led to the further description of numerous other species, expanding the entire group of methanogens [45,46,47].

In 1977, Woese and Fox published the seminal paper describing the *Archaebacteria*, now referred to as *Archaea*, which raised quite a commotion, as the third domain of life was established [48]. The study of the 16S rRNA structure was the key element for the distinction between *Archaea* and *Bacteria*. At first, the difference was found in methanogens, and later confirmed in halophilic archaea and extremely thermophilic archaea. More features of archaea were noticed to have different characteristics, e.g., cell wall, cytoplasmatic membrane, resistance to antibiotics and toxicity of the diphtheria toxin [49,50,51,52,53]. The existence of the third domain of life was supported by this evidence and led to further discoveries related to studies on the metabolism of these microorganisms, protein structure and function and their evolution. It also raised an interest in the cultivation and description of new species that might belong to the domain *Archaea* [54,55,56,57,58].

## 3. Anaerobic Cultivation Techniques

### 3.1. Laboratory Equipment for Cultivation of Methanogens

Anaerobic cultivation differs from aerobic cultivation in many aspects; one of these concerns the instruments with which the laboratory should be equipped. These instruments were established as novel methods of anaerobic cultivation to be introduced. The purpose of maintaining an anaerobic atmosphere and avoiding exposure to O_2_ led to a replacement of some classical instruments and cultivation vessels with the new ones.

The most commonly used cultivation vessels are Hungate or Balch tubes and serum bottles (Figure 2a–c). A Hungate tube is closed with a butyl stopper and a screw cap. Balch tubes and serum bottles are closed with butyl stoppers and sealed with aluminium seals by crimping. This allows the cultivation vessel to be pressurised with substrate gases [59]. However, it also allows the cultivation of gas-producing microbes. In the case of cultivation of, e.g., hydrogenotrophic, autotrophic methanogens, repeated gassing is still necessary because the gaseous substrate in the headspace is consumed according to the specific gas uptake rate and the biomass concentration of gas-utilizing microbes. In the case of thermophilic and hyperthermophilic methanogens, the headspace replenishment must be performed at higher frequency compared to mesophilic or psychrophilic methanogens. Cultivation in larger volumes can be performed using a modified bottle with a neck that can be sealed with a stopper and aluminium cap (Figure 2d,f). The bottle can consist of more than one opening for various purposes [16,60,61,62]. It is also possible to use a pressure-resistant Schott bottle for anaerobic cultivation, which is closed with a GL45 screw cap with an opening and the appropriate butyl rubber stopper (Figure 2e).

The use of anoxic gas or a gas mixture is necessary in anaerobic cultivation. To ensure that there is no O_2_ present in the gas, a copper column was used as another innovation to remove O_2_ from the used gas [14]. The tube filled with copper turnings or copper pellets is heated while the gas flows through the column. When the apparatus is constructed using copper turnings, the electric heating needs to exceed 350 °C. The copper pellets proved to be more effective, as a lower temperature of around 150 °C was sufficient to remove trace amounts of O_2_ [63]. Before introducing H_2_, the column must be absolutely free of O_2_, so it is gassed with N_2_ or argon beforehand. Today, when gasses of purity 5.0 (99.999%) are used in the laboratory, the removal of O_2_ by heated copper columns is no longer necessary.

For the purpose of quantitative gassing, the gassing manifold was invented [60,61,64]. The gassing manifold is necessary when a large number of media or cultures need to be gassed. This is because the gassing of cultures or anaerobization of media is a very time-consuming step. The gassing manifold divides the gas stream into several parallel streams, so that parallel gassing is possible [64]. The gassing manifold can be assembled from individual parts, or it is possible to buy prefabricated gassing manifold (Ochs Glasgerätebau, Bovenden, Germany). These manifolds are often operated manually. However, an electric gassing manifold is also available, e.g., Deoxidized Gas Pressure Injector; IP-8; Sanshin Industrial, Japan. The tubing exiting the gassing manifold can be terminated with a glass syringe containing sterile cotton or a Luer-Lock connector. In the latter case, a sterile filter is inserted between the tubing and the needles. Due to these settings, it is possible to connect the needle through the lock and manipulate closed vessels without exposing them to O_2_. Since all operations with cultures are also performed using syringes and needles, it is also recommended to flush the syringe with the N_2_ or CO_2_ gas before taking a sample of the culture or solutions [14,15].

Some assignments require a strictly anaerobic environment for manipulation, and it is complicated or even impossible to perform them without the use of the anaerobic glove box (Figure 3). It should be noted that the anaerobic glove box is an anoxic environment, but not a sterile one. Glove boxes with built-in HEPA filters are available (Labconco corporation, Kansas City, MO, USA), or a HEPA filter system could be added additionally (Coy Laboratory Products, Grass Lake, MI, USA).

### 3.2. Preparation of Anoxic Media for Methanogens

#### 3.2.1. Composition of Medium for Methanogens

The preparation of media for methanogens can be a very time-consuming process, as many steps are required to make the media suitable for their growth. In addition, the media should reflect the growth conditions of each species in the natural environment, adapting it to the specific requirements of the species for laboratory experiments or for later biotechnological application.

The basis of the media is a mineral buffer solution. The salts in the media can be chlorides or sulphates, although sulphates increase the possibility of SRB growing in the media and competing with methanogens in enrichment cultures [65]. Another important component is the trace element solution, which contains important cofactors of metabolic enzymes. The most used solutions were described by Pfennig and often used by Widdel in his works with phototrophs and sulphate reducers, e.g., SL6 or SL10 [66,67,68,69] (Table 1), or Wolfe’s mineral solutions, made especially for methanogens [60]. Recently, 80 methanogens were grown in 22 chemically defined and/or complex media to assess their methane production and growth characteristics for the identification of high-performance methanogens. The analysis also included multivariate statistical analysis of the medium’s constituents [8].

Vitamin requirements differ among species. Many methanogens can be grown without the addition of vitamins [8,56,70,71], but previous studies have reported that the presence of vitamins supports their growth [72,73]. Other species might be fastidious and require a whole range of different vitamins, e.g., *Methanimicrococcus blatticola* [72,74,75]. Since vitamins are often thermolabile, their filter-sterilized stock solutions are added to media after the sterilization of a basic solution. Similarly to trace elements, complete solutions of vitamins are often prepared in advance and used in various media, e.g., Wolfe’s [60] or Widdel’s [69] vitamin solutions (Table 2).

The use of reducing agents in the media to keep the ORP within a specific range is required for the successful cultivation of anaerobic organisms, not only methanogens [76]. For methanogens, the ORP should not exceed -330 mV. Sodium sulphide (Na_2_S), sodium thioglycolate, dithiothreitol, L-cysteine or sodium dithionite could be used as a reducing agent, which should be added to the media prior to inoculation [76]. The most used reducing agent is Na_2_S. On the other hand, L-cysteine is also used as a source of sulphur for microorganisms instead of a pure reducing agent because an ORP of -340 mV cannot be achieved. The ORP is then set by the final addition of the reducing agent to the medium. If a redox indicator is present in the medium, the coloration of the medium will indicate whether the ORP is within the intended boundary. There are a number of different indicators with resazurin being the most commonly used [76]. However, when methanogens are frequently cultivated in the laboratory, redox indicators are not applied any more [8,64].

Since the organisms are adapted to live in a complex environment in the presence of various microorganisms, this may lead to the necessity of a component that is specific to particular environment and cannot be substituted in vitro. For the successful cultivation of these auxotrophic microorganisms, which initially appear unculturable, additional enhancement of the medium is required. The additive could be yeast extract, ruminal fluid, or sludge fluid [77,78,79]. Media containing carbonate buffer ensure that the pH remains stable. This is quite inconvenient when a wide pH range has to be measured and the optimal pH of the species determined. Different buffer systems may have to be tested.

This is the basic characterization of media for methanogens. Nowadays, there are different media depending on the species or environment of the microorganism of interest, which are freely available on the websites of microorganism collections such as the Deutsche Sammlung für Mikroorganismen und Zellkulturen (DSMZ) or the American Type Culture Collection (ATCC). Unfortunately, the recipes are updated without historical documentation of their former versions, which makes their use somewhat unreproducible. Moreover, the medium recipes are sometimes written in an unnecessarily complicated manner, especially when it comes to the recipe for the cultivation of methanogens. Therefore, it is recommended to assess up-to-date literature in order to select the appropriate medium. In addition, an assessment of growth characteristics of methanogens on different media was recently performed, which could help to select a medium for the cultivation of methanogens [8].

#### 3.2.2. Process of Anaerobization of the Medium

Hungate’s technique (1969) for the preparation of O_2_-free media depends on the heat stability of the solution. Heat-stable media undergo a boiling process in which the dissolved O_2_ is expelled from the media. Heat-labile solutions should be purged with gas bubbles for about 30 min to one hour to remove residual O_2_.

Dispersal of the media can be performed prior to sterilization, which is recommended [14,64], or after sterilization, which carries the risk of contamination [80]. Previously, Hungate’s method of pipetting the media into the vessels while simultaneously gassing the media and the final vessel was quite difficult to manage, as one person must hold the pipette while orally pipetting the media and hold the Hungate tube at the same time (Figure 4a). The vessel is then closed with butyl stopper. A detailed description of this method of media preparation with slight modifications is also provided by Bryant [81]. Not only is the procedure complicated and difficult to perform, but oral pipetting is also prohibited nowadays. Later, apparatus to simplify the process of dispersing was invented, which can be used in the preparation of non-pressurised media (Figure 4b) [63].

Balch proposed to prepare the media in an anaerobic box, which simplified the whole pipetting process. The boiled medium is purged with a gas mixture of N_2_/CO_2_ (4:1 (*v*/*v*)), then the reducing agent is added, the medium is transferred into the anaerobic box and distributed into the cultivation vessels without being exposed to air. In the final step, the closed media are removed from the anaerobic box, the gas atmosphere is replaced, and the media are autoclaved [60]. In this way, the anoxic conditions are maintained and a better adjustment of pH is achieved, as it is carried out after the addition of Na_2_S. 

Another modification as well as the first use of serum bottles in cultivation was devised by Miller and Wolin in 1974. The medium is prepared, then boiled and purged with O_2_-free gas while another stock solution is added. Afterwards, the medium is dispersed into serum bottles that are constantly purged with O_2_-free gas and closed with a butyl robber stopper while still being gassed. In this way, the O_2_ cannot enter the vessel. The media are sterilized by autoclaving.

When cultivating strictly anaerobic SRB and phototrophs, Widdel had also developed methodological variations and procedures. Since these microorganisms do not require a gas atmosphere, the apparatus was designed to fill the entire culturing vessel. The purpose of his designs is to autoclave the medium in the vessel simultaneously with the dispersing apparatus. After autoclaving, the medium is purged with gas while being stirred and dispersed into the culture vessels. The first model was a bottle with a butyl cap for gas purging and with an opening for dispersing the medium. Another model consisted of a conical vessel with a butyl cap, holes in the stopper for thermolabile solution addition, a gassing tube and a dispersing tube. During the process of dispersing, the vessel is turned upside down [69].

The final design is the Widdel flask (Figure 5). The reverse conical shape with a flat base ensures efficient stirring and the possibility of dispersing media from the whole volume of the flask. The individual openings for inserting and taking samples reduce the risk of contamination. At first, the vessel is filled with mineral solution and autoclaved. Then, it is continuously purged with the necessary gas or gas mixture while cooling, and the thermolabile components and reducing agents are added to the media. Due to the pressure created inside through gassing, the medium is poured through the dispensing apparatus into sterile vessels and closed. This method is suitable for media containing thermolabile components or for the media where the precipitation of salts during autoclaving occurs. The disadvantage of this method could be considered the contact with air during dispensing of the media or possible microbial contamination. If this method is used for cultivation of microorganisms that require a headspace, it is difficult to pour the exact volume of media and ensure anoxicity due to the filling mechanism, but it is not a problem to purge the culturing vessels with gas before and immediately after filling the media. Manipulation with this vessel is also described in detail elsewhere [82]. For the purpose of the cultivation of methanogens, this method of media preparation was used in the work on the isolation of *Methanobacterium aarhusense* [83].

In 2010, Wolfe published a method for the preparation of anoxic solutions and media in small volumes. The main advantage of this method is the speed that leads to anoxic solutions. Using the classic purging-with-gas method would require more time. The combination of vacuum–gas cycling and vortexing the entire system to increase the surface area of the liquid effectively expels O_2_ from the culturing vessel [84]. However, even without boiling or vortexing, vacuum–gas cycling prior to autoclaving has been proven efficient for media preparation and cultivation and is used as a proper method nowadays [85,86]. The number of cycles before autoclaving can range from three [85,86] to five [64]. The flushing of the aerobically dispersed media before autoclaving is also efficient [75].

To briefly summarize, the anaerobization of the media could be accomplished via different approaches depending on the objective of the media. The easiest and the most progressive, as well as the most suitable for manipulation and the least time-consuming, is the vacuum–gas cycling method combined with gassing manifold. This method does not require an anaerobic glove box, although a vacuum pump is required. Without a vacuum pump, flushing the media is also an efficient possibility, though slightly more time-consuming. For complicated media consisting of more solutions, the Widdel flask is still a present option that would save material such as needles, syringes and butyl stoppers, which could be destroyed by repeated piercing. These variants can substitute for an anaerobic glove box, which is not present in every laboratory; the manipulation is more difficult or which could be reserved for different tasks.

### 3.3. Cultivation and Pure Cultures Isolation Techniques

Using solid media for strictly anaerobic microorganisms and isolation of pure cultures may be seen as the most difficult part of anaerobic microbiology and biotechnology. The main obstacles are the slow growth rate of some methanogens and maintaining an anaerobic atmosphere. During cultivation, the agar could dry out quickly before the colonies become visible. Not every species is cultivable on solid media, and the higher temperature cannot be achieved with agar, so alternatives, e.g., Gellan Gum, have to be used [63]. Moreover, Petri dishes are not airtight. These are the reasons why anaerobes are mostly cultivated in liquid media.

#### 3.3.1. Petri Dishes Cultivation in Anaerobic Jar

While cultivating anaerobic microorganisms on agar in a Petri dish, it is possible to incubate the plates in the anaerobic glove box. The atmosphere here is provided by the gas from the glove box, but it does not contain a sufficient amount of H_2_ if methanogens are to be cultivated, as the headspace gas contains up to 4.5 Vol.-% H_2_ in N_2_ (forming gas). The low partial pressure of H_2_ could slow down the generation time of methanogens. Anaerobic jar/pressure cylinders can be used to incubate inoculated Petri dishes inside (Figure 6). Some jars are equipped with gassing inlets to fill the headspace with substrate gas. The other option is to use a palladium catalyst inside the jar that uses H_2_ to reduce O_2_. It should be noted that palladium is deactivated by H_2_S, so the catalyst must be replaced before every experiment [87]. There are several commercially available anaerobic pack systems (e.g., AnaeroPouch-Anaero; Mitsubishi Gas Chemical Company, Japan) that have been used and described as support systems for anaerobic cultivation [88]. Some hydrogenotrophic, autotrophic methanogens could be cultivated on agar plates without a H_2_ atmosphere by using, for example, formate as a substrate [85].

When cultivating methanogens on agar, in addition to all the above-mentioned aspects, it is also essential to pay attention to the used agar, its type and brand, its concentration and the amount of Na_2_S [89]. The agar should be washed several times before use to remove all impurities. All factors could affect the morphology of the colonies and the specific growth rate. It is preferable to perform inoculation into deep agar or layered agar [90,91,92].

Special attention should also be paid to the material from which the anaerobic jar is constructed. For long-term cultivations, plastic materials do not have to be O_2_-tight. For this reason, even anaerobic jars with a set atmosphere should also be better kept in an anaerobic box if they are not made of metal [63]. The use of a modified canning jar instead of an anaerobic jar has also been described [90]. The agar plates have been used to isolate *Methanocaldococcus jannaschii* [93] or *Methanothermococcus thermolithotrophicus* [94].

#### 3.3.2. Hungate’s Roll Tube Technique

The most used technique for obtaining pure cultures of methanogens is Hungate’s roll tube technique [14,95,96,97]. This technique is based on the formation of a thin layer of inoculated agar, adhering to the walls of the Hungate tube. The gas phase is anaerobic. During growth, colonies become visible on the agar and as subsurface colonies. These are picked and further dispersed in the new dilution series or cultivated. Bryant preferred to inoculate the picked colonies into a slant medium rather than a liquid medium. The concentrated inoculum was found to be better adapted to the conditions in the new environment, as inoculation into liquid media is sometimes unsuccessful [32,81,98]. Applying this method, the microorganisms grow through the agar and form surface and subsurface colonies. Due to this effect, the morphology of the colonies could differ and that could be wrongly considered as a mixed culture [14]. It is also possible to apply thin layer of agar on the walls and inoculate the roll-tube after the solidification, as it was carried out during the isolation of *Methanogenium marinum* [99]. The purpose of the later inoculation was to avoid exposing the microorganisms to temperatures higher than 15 °C.

A variation of this technique was performed in 1973 by Miller and Wolin with the use of serum bottles instead of Hungate tubes. This method proved to be efficient for the isolation of anaerobic microorganisms such as *Selenomonas ruminantium*, *Ruminococcus albus* or *Methanobrevibacter ruminantium* [16].

#### 3.3.3. Agar Shake Dilution Tube Method

Before the Hungate roll-tube method was developed, the agar shake dilution tube method was used for the cultivation of anaerobic microorganisms [4,27]. The latter method differs from Hungate’s method in the use of the column of semisolid agar in the tube, closed by a butyl stopper. During the growth of methanogens, visible colonies form inside the column. When the methanogens consume a liquid substrate (acetate or methylated compounds), or the technique is used for gas-producing anaerobes, such as Clostridia, they produce an excess volume of gas and the agar in the column is ruptured by the gas bubbles (Figure 7b) [100]. If there are facultatively anaerobic microorganisms present, they grow at the interface with the gas and consume the traces of O_2_, so the tube is completely anaerobic.

During the preparation of these agar shake tubes as well as the preparation of Hungate roll tubes, the temperature of agar must be optimal, as it should still be liquid but not too hot for microorganisms. For this method, modifications also have been introduced, changing the shape of the test tube slightly [101]. This method has been repeatedly used to isolate SRB [80,102] and has also been proven as a method for testing the antibiotic susceptibility of anaerobic microorganisms [103]. Even Hungate initially used the method [33] and it has also been repeatedly used for pure colony isolation in rather more recent publications [83,100], or in serum bottle modification [75].

#### 3.3.4. Lee Tube Method

Another interesting method for the enumeration and cultivation of anaerobic microorganisms is the use of the Lee tube (Figure 8a) [104]. Colonies grow in the reduced agar present in the layer between the outer and inner tubes. The original Lee tube is made of a single piece of glass and sealed with a screw cap, which is more appropriate; a modified Lee tube consisting of two pieces and sealed with a cotton plug, which is not suitable for methanogens using a gaseous substrate, has also been described (Figure 8b) [105]. The Lee tube method has been used repeatedly to isolate and characterise anaerobic bacteria [106] and to maintain the cultures [107], but has not yet been used for methanogens. 

#### 3.3.5. Hermann’s Flat Flask Method

As mentioned above, cultivation on a Petri dish is not convenient due to evaporation of the water and drying of the agar. In 1986, Hermann published a method for anaerobic cultivation on poured agar in a flat flask used originally for phototrophic cultivation (Figure 8c) [108]. Later, in 1992, the method was modified by Olson. An additional opening was added that can be closed and sealed with aluminium. The second opening is used to flush the bottle with anoxic gas during manipulation and inoculation (Figure 8d). Both openings are then sealed and the flask is cultivated [109]. The advantage of this method is that the presence of an anaerobic box is not necessary, the manipulation could be carried out on a laboratory bench and the individual bottles can be checked separately, in comparison to the cultivation of Petri dishes in an anaerobic jar. This method was used, for example, for cultivation of *Methanobrevibacter millerae* and *Methanobrevibacter olleyae* [79].

#### 3.3.6. Single Cell Isolation Methods

For the isolation of strict anaerobes, micromanipulation and single-cell cultivation is also a possible solution used in anaerobic cultivation. The purpose of this method is to take a single cell from the sample and inject it into the media. The isolation of the cell could be performed manually using the microinjectors BactoTip [110] or by laser microscope where the laser beam isolates the cell [111]. For some anaerobic microorganisms, this could be quite time consuming, as they are known for a longer generation time [112,113] and the cultivation of the extracted cells sometimes proves unsuccessful. The recovery rate has been studied and optimization of the technique for different common bacteria was recently accomplished [114]. This method proved successful not only for the isolation of the methanogen *Methanobrevibacter* sp. from termite gut [110], but also for extremophilic archaea such as *Metallosphaera sedulla* and *Saccharolobus solfataricus* [115] or hyperthermophilic archaea [116].

#### 3.3.7. Dilution to Extinction Method

In case the microorganisms are fastidious and cannot be isolated by agar techniques, one option is dilution to extinction. The technique is based on serial dilutions in liquid media, ensuring the loss of most cells until the desired cells are the only present. This method is widely used, although it requires number of cultivation vessels to perform the method [71,117,118]. To remove contaminating microorganisms from an enrichment culture containing methanogens, the addition of appropriate antibiotics, such as ampicillin, vancomycin, clindamycin or kanamycin, could be used to simplify the process of purification [54,66,119]. To exclude fungal contamination, amphotericin B could also be added for the isolation of methanogens from the intestinal sphere [120,121].

### 3.4. Novel Insights in Cultivation Techniques

The anaerobic cultivation methods described by Hungate and their variants are still in use today and are widely applied in anaerobic microbiology and biotechnology. However, there are still many obstacles, and the cultivation of anaerobes is still considered to be very difficult and not always successful, even by following procedures step by step. In order to improve cultivation methods, reduce the difficulty level, increase experimental results and expand the application possibilities, the methods need to be further adapted to today’s requirements and novel devices need to be used for automation, although the improvement of the techniques has not been an issue in the last two decades.

#### 3.4.1. The Six-Well Method

One of the novel methods for cultivation of strictly anaerobic microorganisms was published in 2011 by Nakamura et al. The six-well method using the AnaeroPack system was found to be efficient for the cultivation of a number of methanogens, SRB and syntrophic bacteria [88]. The AnaeroPack system is based on the use of special catalyst sachets without the presence of water or H_2_ production. In the work of Nakamura, inoculated plates in aerobic and anaerobic conditions were placed into a bag with the AnaeroPack system and the gas was recharged with H_2_/CO_2_ (4:1 (*v*/*v*)) mixture. Both variants proved to be successful, although anaerobic conditions for inoculation showed better results.

#### 3.4.2. Growth in Syntrophic Communities

A completely different approach to anaerobic cultivation was published in 2009 by Sakai. There are a number of microorganisms that cannot be cultured or require specific conditions. Most methanogens are cultivated under an atmosphere of H_2_/CO_2_ (4:1 (*v*/*v*)) and an overpressure of 1 to 2 bar. The natural environment can provide even higher pressure and H_2_ concentrations, but there are also environments that have a low H_2_ concentration in which methanogens are also encountered. They often require a lower H_2_ concentration as they are accustomed to grow dependently on the H_2_-producing bacterial species, forming a syntrophic relationship between the two species. It is difficult to simulate the complex relationship in a laboratory. Co-cultivation is the key to successful cultivation and obtaining new species. Instead of adding direct substrate, pre-substrate such as ethanol, butyrate or propionate is consumed by bacteria and H_2_ is gradually released, which is then utilized by hydrogenotrophic, autotrophic methanogens. This method results in the capture of a wider range of methanogens, while the high H_2_ concentration tends to target fast-growing species [122]. The obstacle in the method is connected with the follow-up purification of the culture. Sometimes it is enough to apply the techniques mentioned above and cultivate the microorganism under conditions of higher H_2_ concentration using the liquid dilution method [123] or deep agar dilution tubes [124,125], but the purification is not always accomplished and the whole process is very time-consuming.

#### 3.4.3. Microplate Reader Technique

One of the obstacles to anaerobic cultivation is the testing of chemical substances. In aerobic cultivation, it is possible to obtain results using microtiter plates for which different conditions can be set in one run. Several devices are designed for the continuous measurement of optical density during growth, e.g., Bioscreen C (Oy Growth Curves Ab Ltd., Helsinki, Finland) or Sunrise (Tecan, Männedorf, Switzerland) [126]. This method could be adapted to anaerobic conditions with the use of an oil layer to ensure a microaerobic environment [127], or some of the instruments may provide a Gas Control Module for changing the atmosphere composition. However, strictly anaerobic microorganisms are not able to grow using these methods because the oil layer is not absolutely gas-impermeable, or the gas composition does not favour the growth of methanogens. The third option is to place the entire device inside an anaerobic box to anaerobically measure the optical density continuously or just for a one-time measurement [128,129]. 

For quantitative experiments, a great amount of serum vials or Hungate tubes must be prepared, which is time-consuming and material-heavy. For this reason, various scientific groups have taken an interest in the microplate reading technique to enable large-scale experiments with methanogens. In 2012, Bang et al. tested antimicrobial peptides on methanogens. The microplate was prepared anaerobically in an anaerobic box, cultivated in an anaerobic jar under an atmosphere of H_2_/CO_2_ (4:1 (*v*/*v*)) and then read with a device placed in the anaerobic box to maintain anaerobic conditions [130]. This procedure resulted in the cultivation of hydrogenotrophs and the measurement of their susceptibility to various antimicrobial peptides. A similar experiment was performed to test chemical substances using a microtiter plate, but with the differences that the microtiter plate was cultivated directly in an anaerobic box and the microorganisms were growing on a substrate other than H_2_ [131].

#### 3.4.4. Microfluidic Techniques

In recent years, microfluidics has achieved more recognition as an efficient and high-throughput method for cultivation and testing different conditions. It is obvious that its applications are broad, as it offers a variety of conditions in a small space. Applications of microfluidics in anaerobic conditions are focused primary on human gut microbiota and capturing species that are not yet culturable [132,133,134]. This technology was used to determine the optimal conditions for the growth of *Methanosaeta concilii*, using an anaerobic microbioreactor under a N_2_/CO_2_ atmosphere [135]. However, the aforementioned experiments were conducted in an anaerobic chamber, and hydrogenotrophic, autotrophic methanogenesis has not yet been investigated.

At the borderline between microfluidics and the classic microtiter plate method is a device designed for anaerobic cultivation in a 48-well plate (e.g., BioLector, m2p-labs GmbH, Baesweiler, Germany). It allows measurements of optical density, pH, biomass and fluorescence, as well as continuous feeding of the reactors. The instrument has been tested for fermentation of e.g., *Clostridium* sp. and *Lactobacillus* sp. [136], although the potential use for strict anaerobes such as methanogens has not yet been tested.

## 4. Quantification Techniques

During the cultivation of microorganisms, it is necessary to measure their growth, use these data to calculate growth parameters to define these microorganisms, and compare different growth conditions. Knowledge of the growth kinetics is also important for biotechnological applications, to relate the substrate conversion or product formation to biomass, or to directly calculate gas production using pure methanogenic cultures. Methods for quantification of growth of anaerobic microorganisms were recently reviewed by Mauerhofer et al. [76]. These quantification methods are non-growth-dependent.

### 4.1. Optical Density Measurement Technique

Optical density measurement is still a widely used and classic method to measure biomass and calculate specific growth rate (Equation (2)) [137]:(2)$$\mu =\frac{\ln_{{\mathrm{OD}}_{2}}{-\ln}_{{\mathrm{OD}}_{1}}}{t_{2}{-t}_{1}}$$
where µ is the specific growth rate (h^−1^), OD_1_ and OD_2_ are the optical density values determined at the respective times t_1_ and t_2_ (h).

Optical density of methanogens is often measured at wavelength 578 nm [138,139,140] but it could be also measured at wavelength 600 or 660 [74,141]. The measured optical density has to be multiplied by a conversion factor, which is specific to every organism and must be experimentally determined, to obtain data for biomass or number of cells per volume (Equation (3)):(3)$$\frac{x}{V}=a\cdot \mathrm{OD}$$
where x represents the number of cells in the sample, V is the volume of the sample (mL or L), OD is optical density and *a* is the conversion factor.

In some cases, this method cannot be used due to cell aggregation, the presence of precipitates or possible interference of resazurin in the medium. For the latter complication, it is recommended to add a reduction agent before measurement, which results in decolorization of the medium, or even better to omit resazurin addition into the media at all if OD measurement is considered.

### 4.2. ATP Determination Method

When optical density measurement is not an option for biomass quantification, there are other possible methods, such as the calculation based on direct cell counting [66,71] or adenosine triphosphate (ATP) determination. ATP determination could be also used to determine the growth of the culture [142,143]. Nowadays, there are many commercial products for ATP assay available, based on luminescence (luciferin/luciferase system) or colorimetric methods.

### 4.3. Methods Requiring the Cultivation of Anaerobic Microorganism

#### 4.3.1. Methane Production Measurement Techniques

Methanogens have the advantage of not only measuring biomass, but also releasing methane as a metabolic end product. Methane, as the main component of biogas, carries the conserved energy and it is highly important to obtain the biggest energy yield from the different used substrates. The growth rate can be calculated from methane production with the use of Equation (4) [117,144,145]:(4)$$\mu_{{\mathrm{CH}}_{4}}=\frac{\ln\left(M_{2}{/M}_{1}\right)}{t_{2}{-t}_{1}}$$
where µ_CH4_ is the specific growth rate (h^−1^) calculated based on methane production. M_1_ and M_2_ represent the produced methane (mmol) measured at the respective times t_1_ and t_2_ (h).

When the culture’s growth slows down and enters the stationary phase, the optical density does not change rapidly. On the other hand, methane production still occurs at high rate. To capture the methane produced over time, methane evolution rate (MER) (Equation (5)) is used [64,146]:(5)$$\mathrm{MER}=\frac{{\Delta n}_{{\mathrm{CH}}_{4}}}{{(t}_{2}{-t}_{1})\cdot V}$$
where MER is methane evolution rate (mmol L^−1^ h^−1^), Δ*n*_CH4_ is the difference between the produced methane (mmol) measured at the respective culturing times t_1_ and t_2_ (h), and V is the volume of the media (L).

During hydrogenotrophic, autotrophic methanogenesis, the methane is produced according to the molar equation as mentioned above in Equation (1).

Based on the equation, four moles of H_2_ and one mole of CO_2_ is consumed as one mole of methane is produced. Since the substrates are both in the form of gas, it is possible to use the ideal gas law (IGL) (Equation (6)) for calculation the methane concentration in the headspace of the culture vessel:(6)$$n=\frac{p\cdot V}{R\cdot T}$$
where n is amount of the substance of gas (mol), p is pressure (Pa), V is the volume of the headspace (m^3^), R is the universal gas constant (J K^−1^ mol^−1^), and T represents the temperature (K).

For the application of this law, two ways of measurement exist. The first is based on the change in pressure in the headspace, which is measured with a manometer. The pressure drop represents the four moles of consumed H_2_ (Equation (7)):(7)$$n_{{\mathrm{CH}}_{4}}=\frac{{(p}_{0}-p)\cdot V}{R\cdot T\cdot 4}$$
where n_CH4_ is the amount of produced methane in the headspace (mmol), p_0_ represents the initial pressure in the vessel (Pa), p represents the actual pressure in the vessel (Pa), and V is the volume of headspace (L).

The other option is to combine the actual pressure of the vessel to calculate absolute moles of gas in the headspace and the percentual concentration of methane measured by gas chromatography (GC) (Equation (8)):(8)$$n_{{\mathrm{CH}}_{4}}=\frac{p\cdot V}{R\cdot T}.\frac{\;\%\;\mathrm{GC}}{100}$$
where n_CH4_ is the amount of produced methane in the headspace (mmol), p represents the actual pressure in the vessel (Pa), V is the volume of headspace (L), and % GC represents percentual representation of methane in the headspace.

The first way is much faster but depends heavily on the pressurization of the vessels. It is also recommended to flush the headspace with the gas after every measurement to set the same conditions and pressure. This is because gas leaks may occur when measuring with a manometer and the results may be greatly affected during the second measurement. The use of GC is dependent on the actual pressure and the repeated flushing is not required if there is still a significant amount of substrates, respectively H_2_, present in the headspace [64]. The use of GC is also the only option to directly measure methane produced from formate or by methylotrophic or acetoclastic methanogens.

#### 4.3.2. Manometric OxiTop Measurement

Apart from typical reactors, a simple device for measuring pressure in the vessel is the anaerobic OxiTop (OxiTop^®^-IDS, WTW GmbH, Germany). With two openings for manipulation through needles, the substrate can be added or the samples removed. Although it has not been yet directly used on pure cultures, potential application on cultures is possible. Continuous pressure measurement gives insight into gas formation kinetics.

#### 4.3.3. Indirect Quantification of Produced Methane via Weight Gain

Another indirect method to determine produced methane uses the weight increase due to the formation of water during hydrogenotrophic methanogenesis [64]. This method is based on repeated measures of weight in regular intervals (Figure 9). The weight increment represents the two moles of water formed on the one mole of methane and is described as the water evolution rate (WER) (Equation (9)).
(9)$$\mathrm{WER}=\frac{{\Delta m}_{H_{2}O}}{{(t}_{2}{-t}_{1}){.\mathrm{Mr}}_{H_{2}O}.V}$$
where WER is the water evolution rate (mol L^−1^ h^−1^), Δm_H2O_ is the mass difference (g) measured in the respective culturing times t_1_ and t_2_ (h), V is the volume of the media (L), and Mr_H2O_ is the molar mass of water (g mol^−1^).

In combination with the pressure measurement, this approach presents the metabolic activity of hydrogenotrophic, autotrophic methanogens. It is applicable to fast-growing methanogens where the weight increase is easily measurable and noticeable. For slowly growing methanogens, it is recommended to set a longer interval of measurement. In addition, drops of the suspension may leak when handling the serum bottles, leading to inaccuracies in the measurement and a slow decrease in the weight in the control serum bottle [147].

## 5. Up-Scaling Process during Anaerobes Cultivation

Classic use of serum bottles is required for cultivation of species, capturing new species of microorganisms, and studying of material degradation in the primary steps of research. The setup is called closed-batch cultivation. However, the repeated gassing results in unbalanced growth and the maximal pressure of the headspace should not exceed 3 bar. Therefore, cultivation in an environment where the culture could achieve balanced growth according to a growth curve with enough substrate is necessary. Furthermore, in order to simulate processes in industrial reactors, up-scaling is needed to achieve more accurate results. It is also the way to obtain microbial cell mass for further analyses.

### Batch, Fed-Batch and Continuous Cultivation

Bioreactors can be operated in batch, fed-batch, and continuous modes. This division is based on the experimental design and on the level of intervention in the reactor. Batch mode requires the least intervention. The medium, substrate and culture are prepared at the beginning and harvested at the end of the experiment. The fed-batch operation mode differs from the batch system as the carbon and energy source are added during the run of the reactor at a defined rate. This setup is limited by the volume of the reactor as there is no outflow of the excess liquid (in the case of a liquid feed). Continuous cultivation, as the name suggest, depends on the continuous inflow of substrate and nutrients and outflow of suspension and products.

Up-scaling the cultivation of methanogens encounters several different obstacles. Batch cultivation of pure cultures depends on the amount of substrate, which does not have to be sufficient to obtain the requested yield of biomass. The greatest advantage of this setup is the possibility of pressurisation of the reactor and setting conditions for observing the growth of the population. The more the reactor is pressurised, the more substrate is available to the culture and substrate conversion could be observed through pressure measurement [146,148]. In special occasions, the gas could be removed and the reactor repeatedly pressurised.

A fed-batch system ensures continuous substrate addition; in the case of hydrogenotrophic, autotrophic methanogens, this means a flow of H_2_/CO_2_ (4:1 (*v*/*v*)) mixture (Figure 10). Continuous cultivation also removes suspension and ensures a stable medium volume [76]. Reactors with a continuous supply of H_2_/CO_2_ gas mixture are hard to pressurise to increase the solubility of the gases. One way to solve the problem is to stir the suspension to increase the liquid-phase interface. Not every species of methanogen is able to withstand agitation and atmospheric pressure. There were some cases where the continuous reactor was pressurised to an absolute pressure of 1.22 bar [149] or 1.75 bar [150], but this does not reach the pressure achievable in closed batch cultivation, where a pressure of tens of bars could be easily set. There is one exception described of a constructed, pressurised fed-batch reactor, with a pressure 10 to 20 bars, which had been operated with *M. marburgensis* [151]. Due to water formation during hydrogenotrophic, autotrophic methanogenesis, it is also suggested to take a notice of increasing volume of medium during the planning of the experiment. If the expected production is greater, removal of the excess volume and addition of medium should be considered.

In 1968, Bryant et al. successfully cultivated *Methanobacterium* strain M.o.H. in a fed-batch setup with a reactor volume of 12 L and under continuous gassing [152]. To date, the experiments of fed-batch or continuous operation setups of different volumes have been successfully carried out with mainly thermophilic microorganisms, such as *Methanothermobacter* species [138,149,153,154,155,156], or microorganisms from the order *Methanococcales* [138,157]. The exceptions are mesophilic *Methanococcus maripaludis* [158] and the mentioned *Methanobacterium* strain M.o.H [152].

Acetoclastic and methylotrophic species are not dependent on pressure, and it is possible to cultivate them under a H_2_-free atmosphere. In the case of closed batch cultivation, the pressure would increase instead [145].

## 6. Conclusions

Cultivation techniques for anaerobic microorganisms, especially strictly anaerobic ones such as methanogens, vary greatly, from classic methods of aerobic cultivation to modern anaerobic high-throughput methods. Because of the difficulty of dealing with the fastidious anaerobic microorganisms and the required equipment for the cultivation, not many laboratories focus on anaerobic methods and anaerobic microorganisms. With this review, the historical development of culturing methods and the potential applications are presented in order to introduce anaerobic cultivation to the scientific community and to assist in reflecting on these methods to develop a new generation of anaerobic cultivation methods. This review also emphasises the importance of persevering in the knowledge and pitfalls of anaerobic cultivation techniques in anaerobic microbiology and biotechnology.

## Figures and Tables

**Figure 1 microorganisms-10-00412-f001:**
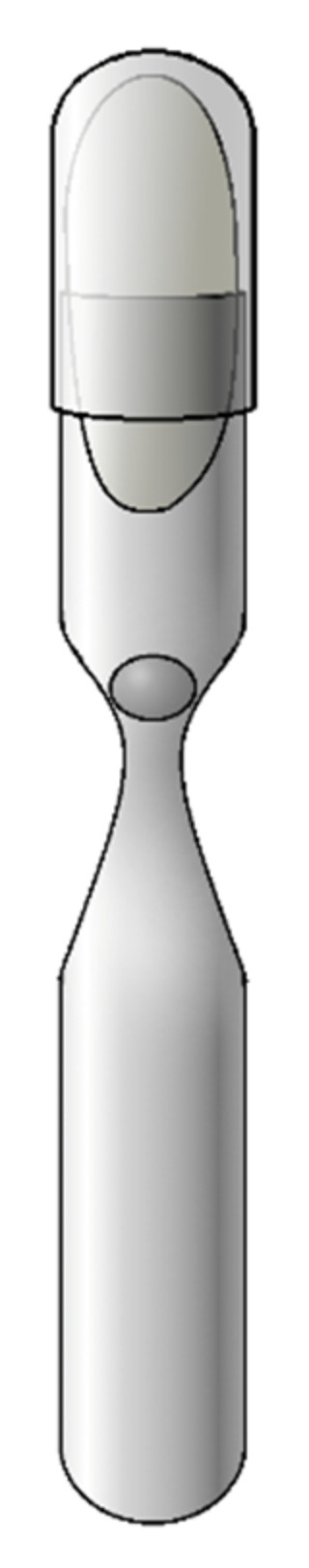
Hall’s constricted tube with marble seal (Hanišáková).

**Figure 2 microorganisms-10-00412-f002:**
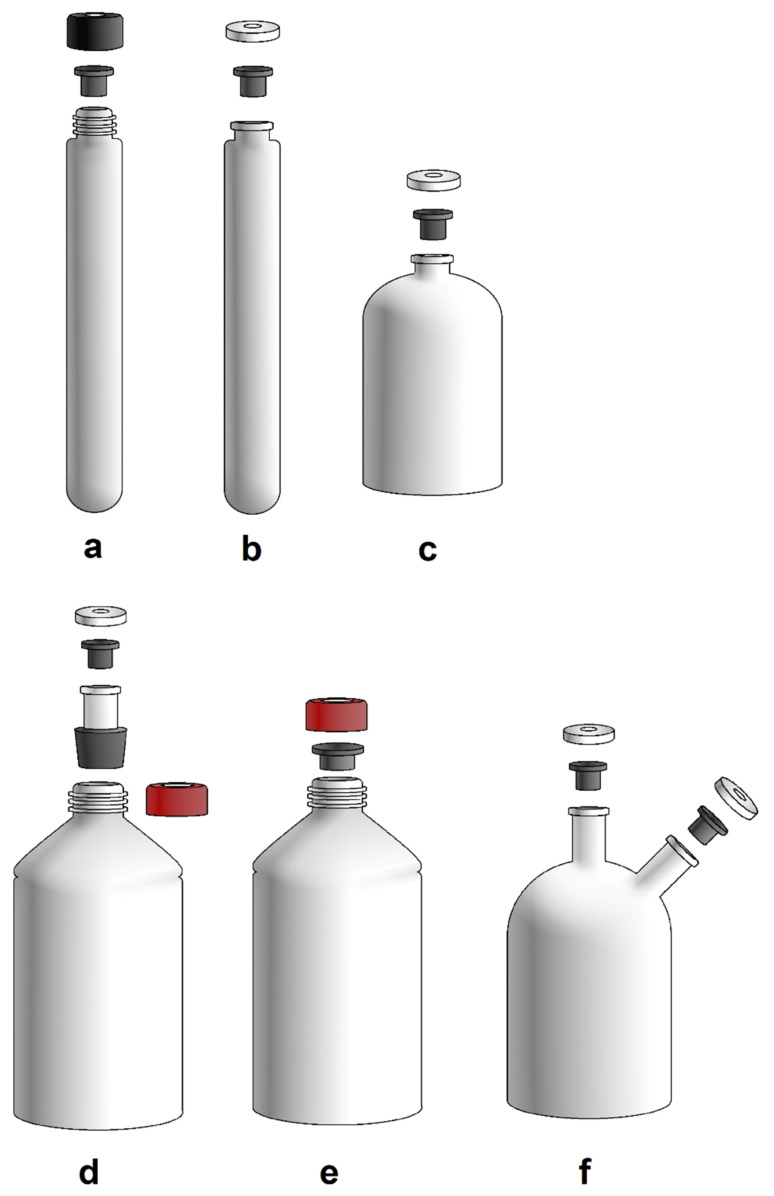
The most used cultivation vessels: (**a**) Hungate tube; (**b**) Balch tube; (**c**) serum bottle; and culturing vessels for greater volume: (**d**) modified bottle with neck (according to Balch, 1979); (**e**) pressure bottle with butyl stopper and GL45 opening; (**f**) modification of bottle with more openings (Hanišáková, according to Miller and Wolin, 1974).

**Figure 3 microorganisms-10-00412-f003:**
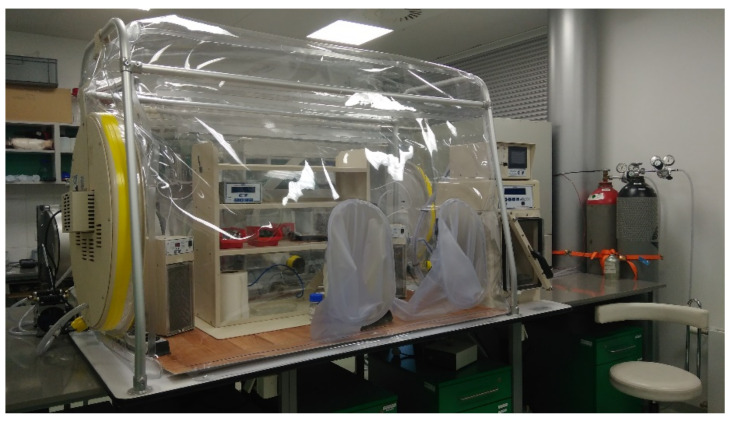
Anaerobic box (Coy Laboratory Products, USA) (Photo: Laboratory).

**Figure 4 microorganisms-10-00412-f004:**
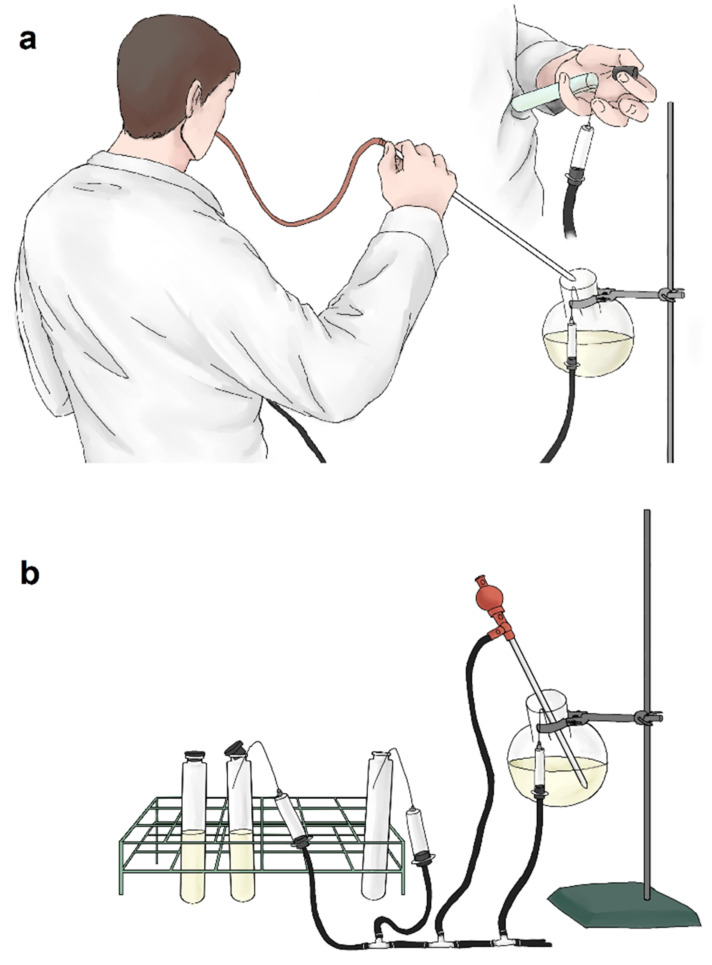
(**a**) Media pipetting described by Hungate and Bryant. (**b**) Dispersing media illustrated according to Sowers (Hanišáková).

**Figure 5 microorganisms-10-00412-f005:**
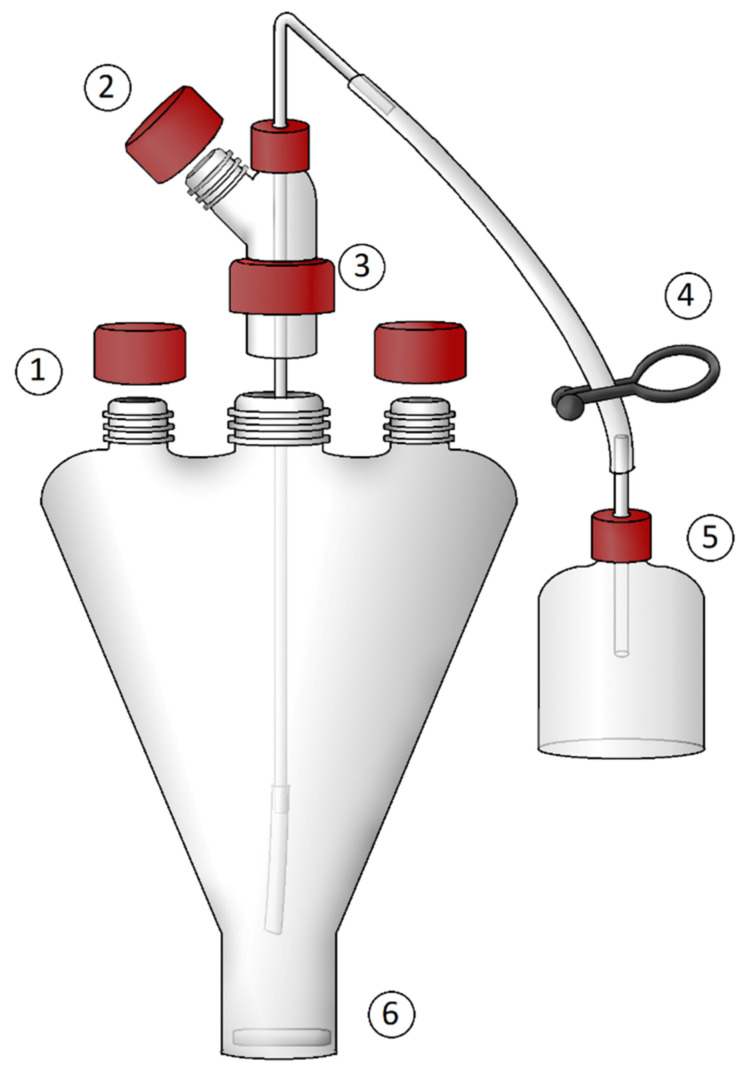
Widdel’s flask. ① Openings for input/output of media samples. ② Opening for gas entry. ③ Central opening for media filling. ④ Clamp for media filling. ⑤ Filling funnel for media. ⑥ Magnetic stirrer (Hanišáková).

**Figure 6 microorganisms-10-00412-f006:**
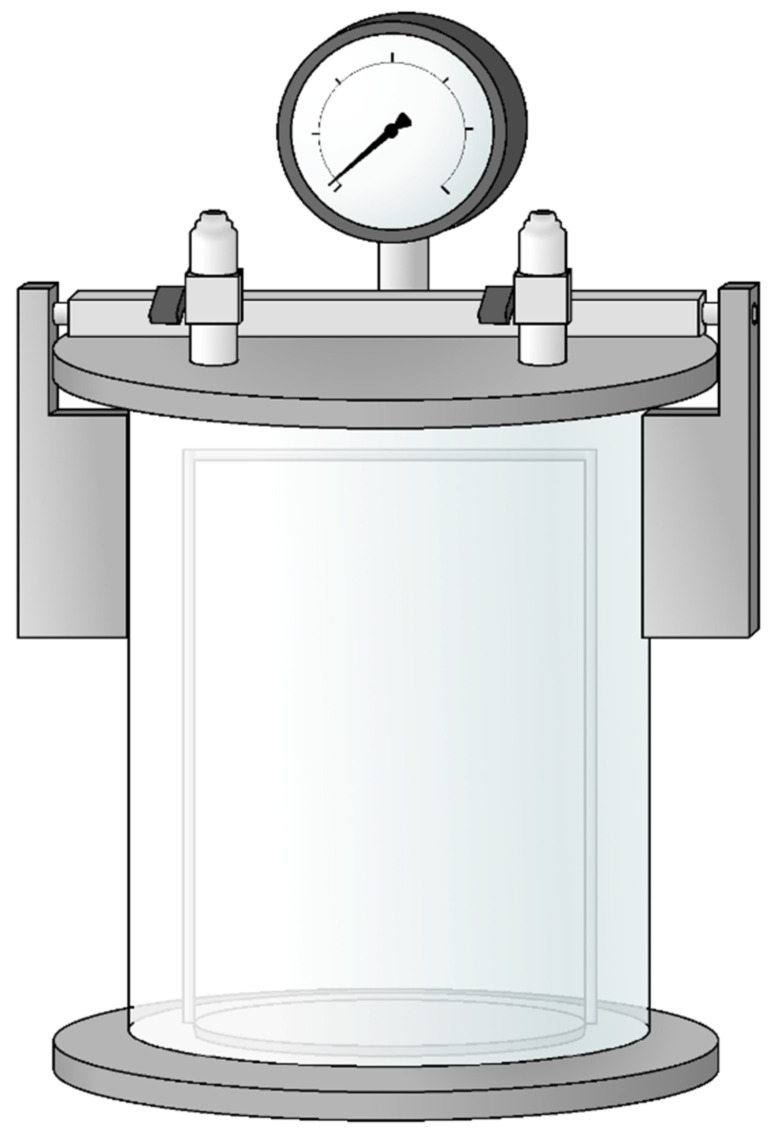
Illustration of anaerobic jar for anaerobic Petri dish cultivation (Hanišáková).

**Figure 7 microorganisms-10-00412-f007:**
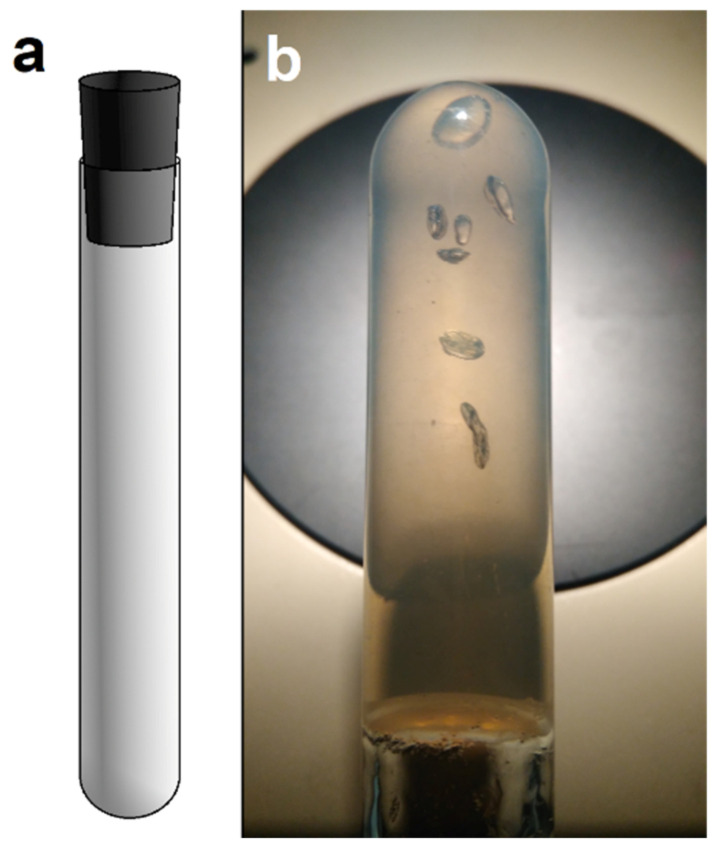
Test tube for agar shake. (**a**) Test tube with butyl stopper. (**b**) Photo of gas bubbles inside the agar (Hanišáková, photo: laboratory).

**Figure 8 microorganisms-10-00412-f008:**
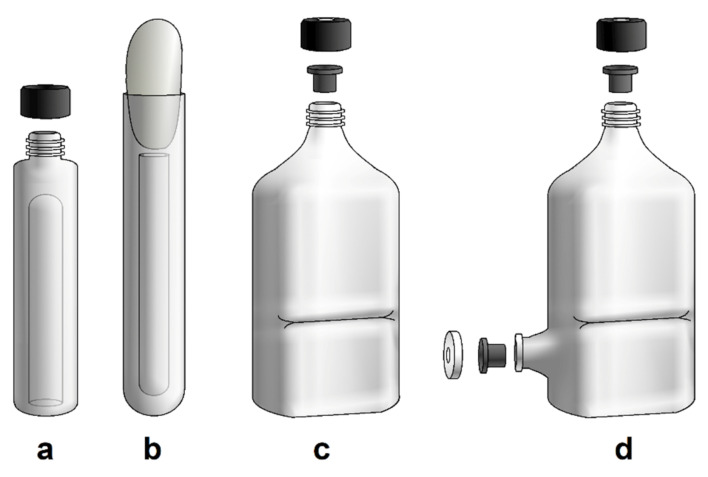
Cultivation vessels for pure colony isolation. (**a**) Lee tube; (**b**) modified Lee tube; (**c**) flat flask — original used by Hermann; (**d**) flat flask modification with additional opening (Hanišáková).

**Figure 9 microorganisms-10-00412-f009:**
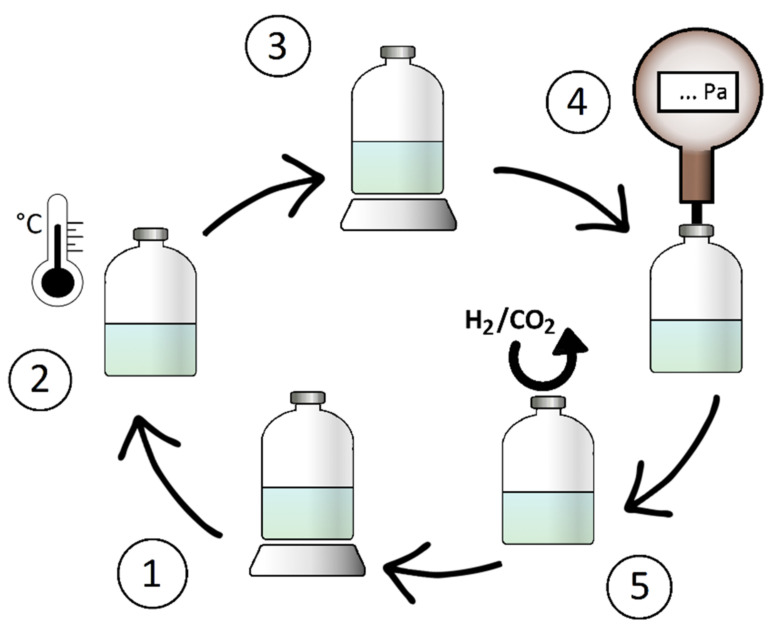
The process of quantification of methane via produced water during hydrogenotrophic methanogenesis. ① The weight of inoculated serum bottle is measured. ② Incubation of the serum bottle. ③ The weight of the serum bottle after incubation is measured. ④ The pressure in the headspace is determined. ⑤ The serum bottle is flushed with gas. (1) The weight of the serum bottle is again measured, presenting the weight difference and gained mass of water (Taubner & Rittmann, 2016, redrawn by Hanišáková).

**Figure 10 microorganisms-10-00412-f010:**
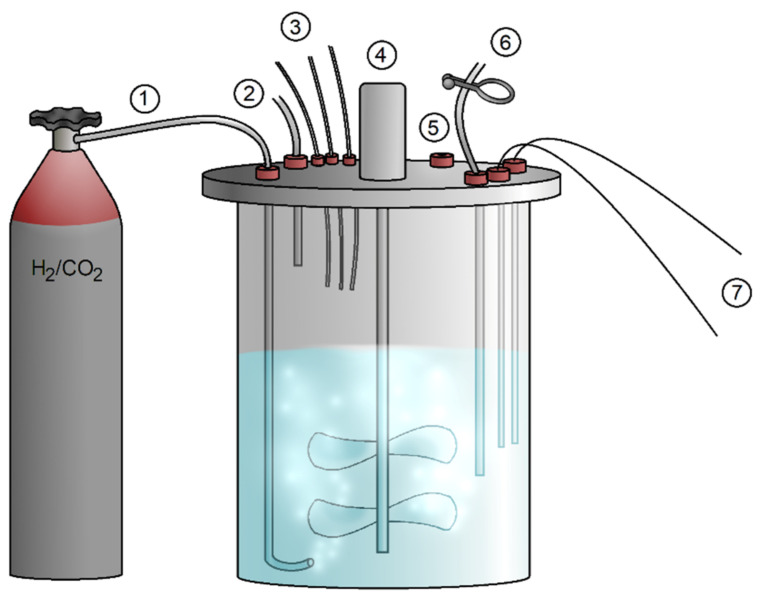
Model of fed-batch bioreactor. ① Gas mixture inflow. ② Gas outflow. ③ Acid-base and sulphide solution input. ④ Stirring mechanism. ⑤ Opening for inoculum input. ⑥ Sample output. ⑦ Temperature sensor and pH electrode (Hanišáková).

**Table 1 microorganisms-10-00412-t001:** Composition of the most common trace element solution used in methanogenic media preparation.

	Wolfe’s Solution ^1^ (g/L)	SL10 ^2^ (g/L)	SL6 (g/L)
Nitrilotriacetic acid (NTA)	1.5	-	-
MgSO_4_.7H_2_O	3	-	-
MnSO_4_.H_2_O	0.5	-	-
MnCl_2_.4H_2_O	-	0.1	0.003
NaCl	1	-	-
NiCl_2_.6H_2_O	-	0.024	0.002
FeSO_4_.7H_2_O	0.1	-	-
FeCl_2_.4H_2_O	-	1.5	-
CoCl_2_.6H_2_O	0.1	0.19	0.02
CaCl_2_	0.1	-	-
ZnSO_4_.7H_2_O	0.1	-	0.01
ZnCl_2_	-	0.07	-
CuSO_4_.5H_2_O	0.01	-	-
CuCl_2_.2H_2_O	-	0.002	0.001
AlK(SO)_4_.12H_2_O	0.01	-	-
H_3_BO_3_	0.01	0.006	0.03
Na_2_MoO_4_.2H_2_O	0.01	0.036	0.003
Reference	[60]	[68]	[67]

^1^ dissolve NTA in 500 ml water and adjust pH to 6.5 with KOH, then add the rest of the compounds. ^2^ dissolve FeCl_2_.4H_2_O in 10 ml 25% HCl, add deionized water and dissolve the rest of the salts. Fill to volume of 1000 mL.

**Table 2 microorganisms-10-00412-t002:** The most common vitamin solution used in methanogenic media preparation.

	Wolfe’s Solution (mg/L)	Widdel’s 5 Vitamin Solution (mg/L)
Pyridoxine-HCl	10	15
Thiamine-HCl	5	-
Riboflavin	5	-
Nicotinic acid	5	10
Calcium pantothenate	5	5
p-Aminobenzoic acid	5	4
α-Lipoic acid	5	-
Biotin	2	1
Folic acid	2	-
Cyanocobalamin	0.1	-
Reference	[60]	[69]

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
