# Peer review of "The Historical Development of Cultivation Techniques for Methanogens and Other Strict Anaerobes and Their Application in Modern Microbiology"

_microorganisms, 2022, doi:10.3390/microorganisms10020412_

Round 1
Reviewer 1 Report
Line 28. This is due to the fact that O2 is toxic, to varying degrees and depending on the microbe, to anaerobic microorganisms. – Please, clarify the sentence.
Line 41 Apart from this, there are laboratory limitations towards many different anaerobic cultivation setups that would enable cultivation of organisms under a wide range of experimental conditions, to use high-throughput methods, requirement towards a particular gas composition and applied pressure, and the toxicity and flammability of some microbial substrates and product gasses, which might render cultivation often difficult to impossible in standard microbiological laboratories. The sentence is to long. I recommend shortening and editing
Line 126. The gas produced during anaerobic cultivation also aroused curiosity, what the composition of the gas is - Please, clarify the sentence.
Line 31. in vitro – Please, use italic style
Line 531 To remove contaminating bacteria from a enrichment culture containing methanogens, the addition of appropriate antibiotics, such as ampicillin, vancomycin, clindamycin or kanamycin, could be used to simplify the process of purification [54,66,119]. - Is the use of an antifungal drug (amphotericin) recommended?
Author Response
Dear Opponent,
thank you for your time and valuable comments on our article. In the attached file we respond to each comment. Thank you for reviewing the corrected version of the manuscript.
Authors
Review Report Form
Reviewer 1
Line 28. This is due to the fact that O2 is toxic, to varying degrees and depending on the microbe, to anaerobic microorganisms. – Please, clarify the sentence.
Reply: Thank you for your comment. The sentence is changed: „This is due to the fact that O2 is toxic to anaerobic microorganisms, to varying degrees and depending on the microbe [1] and on the oxidation-reduction potential (ORP), the optimal value of which differs among anaerobic species.”
Line 41 Apart from this, there are laboratory limitations towards many different anaerobic cultivation setups that would enable cultivation of organisms under a wide range of experimental conditions, to use high-throughput methods, requirement towards a particular gas composition and applied pressure, and the toxicity and flammability of some microbial substrates and product gasses, which might render cultivation often difficult to impossible in standard microbiological laboratories. The sentence is to long. I recommend shortening and editing
Reply: Thank you for your comment. The sentence was edited and splitted into more sentences to be more clear and readable: “Apart from this, there are laboratory limitations towards many different anaerobic cultivation setups with regard to experimental conditions. That means, using high-throughput methods or studying requirement towards a particular gas composition and applied pressure. Also, the toxicity and flammability of some microbial substrates and product gasses alone is an important factor, which might render cultivation often difficult to impossible in standard microbiological laboratories.”
Line 126. The gas produced during anaerobic cultivation also aroused curiosity, what the composition of the gas is - Please, clarify the sentence.
Reply: Thank you for your comment, the section was fixed to be clearer: “During the experiments with fermentation and anaerobic cultivation of mixed samples, the gas was produced, and that aroused curiosity, what the composition of this gas is.”
Line 333. in vitro – Please, use italic style
Reply: Thank you, the text is corrected.
Line 531 To remove contaminating bacteria from an enrichment culture containing methanogens, the addition of appropriate antibiotics, such as ampicillin, vancomycin, clindamycin or kanamycin, could be used to simplify the process of purification [54,66,119]. - Is the use of an antifungal drug (amphotericin) recommended?
Reply: Thank you for your comment. Yes, amphotericin B could be used for omitting fungal contaminations. This information was added to the text: „To exclude fungal contamination, the addition of amphotericin B cloud be also done for the isolation of methanogens from the intestinal sphere [120,121].“
Reviewer 2 Report
The study describes the development of methods for cultivation of strict anaerobic microorganisms. The difficult beginnings of laboratory work with anaerobes, the study of their properties, the first isolation of anaerobes and the discovery of the Archea are described detail. This section is somewhat lengthy on the current scientific text and has little professional contribution.
L.63 - 210 I recommend shortening and editing this part significantly.
Anaerobic cultivation techniques are described in historical order and are supplemented by schematic illustrations of anaerobic cultivation equipment.
To better understand the importance of older methods, it would be appropriate to include an assessment of their contribution to the development of new cultivation approaches and techniques.
The part describing the composition of specific media with the possibilities of their use is clearly processed. In this section, I appreciate the critical assessment of some of the media appearing on the market today.
The overview also includes various methodologies of media anaerobization and attention is paid to the development of methods for the preparation of small volumes of media.
It would be appropriate to supplement the text with a brief evaluation and comparison of individual methods, e.g. assessment of their limitations or benefits, such as technical and time difficulty or cost, etc.
The list of quantification techniques would also deserve an indication of the importance and benefits of their application.
To better understand the importance of quantification techniques, it would be appropriate to supplement the text with a definition of their importance and contribution to biotechnological practice.
L. 696 equation 9 - requires control.
L. 23 Key words - a revision would be appropriate given the content of the article.
Author Response
Dear Opponent,
thank you for your time and valuable comments on our article. In the attached file we respond to each comment. Thank you for reviewing the corrected version of the manuscript.
Authors
Review Report Form
Reviewer 2
The study describes the development of methods for cultivation of strict anaerobic microorganisms. The difficult beginnings of laboratory work with anaerobes, the study of their properties, the first isolation of anaerobes and the discovery of the Archaea are described detail. This section is somewhat lengthy on the current scientific text and has little professional contribution.
L.63 - 210 I recommend shortening and editing this part significantly.
Reply: As this review is focused on historical origin and development of anaerobic cultivation techniques, focused on strict anaerobes, this part describing the process of discovery of anaerobic cultivation may not have contribution to the newest techniques in this field. But the purpose is to remind and describe the findings that could be forgotten, because it is still relevant, and scientist can still benefit from the historical origin and should not forget it.
Anaerobic cultivation techniques are described in historical order and are supplemented by schematic illustrations of anaerobic cultivation equipment.
To better understand the importance of older methods, it would be appropriate to include an assessment of their contribution to the development of new cultivation approaches and techniques.
Reply: Thank you for your comment. Some of the contributions are already mentioned, e.g. Hungate’s method and their modification, which is used till today, or agar shake tube method, which was method used before Hungate’s tube and is also used till today. The biggest contribution of all techniques is that they were used successfully and led to isolation of new species. Also, the advance in these methods is slow and that’s why nowadays there are gaps in fields that are working in aerobic microbiology but not that for strict anaerobes.
The part describing the composition of specific media with the possibilities of their use is clearly processed. In this section, I appreciate the critical assessment of some of the media appearing on the market today.
The overview also includes various methodologies of media anaerobization and attention is paid to the development of methods for the preparation of small volumes of media.
It would be appropriate to supplement the text with a brief evaluation and comparison of individual methods, e.g. assessment of their limitations or benefits, such as technical and time difficulty or cost, etc.
Reply: Thank you for your comment. To the mentioned section, the following part was added: “To briefly summarize, the anaerobization of the media could be done by different approaches depending on the objective of the media. The easiest and the most progressive, as well as suitable for manipulation and the least time-consuming, is the vacuum-gas cycling method combined with gassing manifold. This method does not require anaerobic glove box, although a vacuum pump is required. Without a vacuum pump, flushing the media is also efficient possibility, slightly more time-consuming. But for complicated media consisting of more solutions, Widdel flask is still a present option, that would save material – such as needles, syringes and butyl stoppers, which could be destroyed by repeated pearcing. These variants can substitute anaerobic glove box that is not present in every laboratory, the manipulation is more difficult or could be reserved for different tasks.”
The list of quantification techniques would also deserve an indication of the importance and benefits of their application.
To better understand the importance of quantification techniques, it would be appropriate to supplement the text with a definition of their importance and contribution to biotechnological practice.
Reply: Thank you for your comment. The section of quantification techniques was supplemented with following explanation: “During the cultivation of microorganisms, it is necessary to measure their growth, use these data to calculate growth parameters to define these microorganisms and compare different growth conditions. Knowledge of the growth kinetics is also important for biotechnological applications, to relate the substrate conversion or product formation to biomass, or to directly calculate biogas production using pure methanogenic cultures.”
- 696 equation 9 - requires control.
Reply: Thank you for your comment, the part with equation was corrected.
- 23 Key words - a revision would be appropriate given the content of the article.
Reply: Thank you, the keywords were changed in the text.
Round 2
Reviewer 2 Report
The authors responded to the comments and made some adjustments to the text. I agree to publish this article.